# No-reference panoramic image quality assessment based on multi-region adjacent pixels correlation

**Xinpeng Huang, Xin Liu, Wenxin Ding, Chunli Meng, Ping An** [ID] *

Shanghai Institute for Advanced Communication and Data Science, School of Communication and Information Engineering, Shanghai University, Shanghai, China

* anping@shu.edu.cn

## Abstract

The distortion measurement plays an important role in panoramic image processing. Most measurement algorithms judge the panoramic image quality by means of weighting the quality of the local areas. However, such a calculation fails to globally reflect the quality of the panoramic image. Therefore, the multi-region adjacent pixels correlation (MRAPC) is proposed as the efficient feature for no-reference panoramic images quality assessment in this paper. Specifically, from the perspective of the statistical characteristics, the differences of the adjacent pixels in panoramic image are proved to be highly related to the degree of distortion and independent of image content. Besides, the difference map has limited pixel value range, which can improve the efficiency of quality assessment. Based on these advantages, the proposed MRAPC feature collaborates with the support vector regression to globally predict the quality of panoramic images. Extensive experimental results show that the proposed no-reference panoramic image quality assessment algorithm achieves higher evaluation performance than the existing algorithms.

## 1 Introduction

Compared with normal planar images, panoramic images adopt the image stitching technique and can achieve free viewing in various directions, which provide users an immersive experience with the help of head-mounted displays (HMDs). In practical, the operations, including stitching, blending, projecting and encoding, inevitably lead to quality degradation of panoramic images, and further cause discomfort and physical illness [1]. Therefore, it is urgent for panoramic images to propose an accurate quality assessment algorithm in the future immersive applications.

The existing quality assessment algorithms consist of full-reference image quality assessment methods and no-reference image quality assessment methods [2]. The traditional full-reference image quality assessment methods aim to solve the problem of non-uniform sampling by peak signal to noise ratio (PSNR) or structural similarity (SSIM) [3]. Yu et al. [4] proposed a method to calculate PSNR on a spherical surface, called S-PSNR (spherical PSNR).

**Data Availability Statement:** All relevant data are within the paper.

**Funding:** This work was supported in part by the National Natural Science Foundation of China [Grants 62001279, 62020106011, 62071287,

61901252], and Science and Technology
Commission of Shanghai Municipality [Grant
20DZ2290100]. There was no additional external
funding received for this study.

**Competing interests:** The authors have declared
that no competing interests exist.

Sun et al. [5] proposed the weighted-to-spherically-uniform PSNR (WS-PSNR) that assigns
different weighting maps for different reprojection forms. Zakharchenko et al. [6] presented
craster parabolic projection PSNR (CPP-PSNR) with the least shape distortion. In addition to
PSNR, Zhou et al. [7] proposed weighted-to-spherically-uniform SSIM (WS-SSIM) that is sim-
ilar to WS-PSNR. Mittal et al. [8] proposed a blind/referenceless image spatial quality evaluator
(BRISQUE) that is a widely used no-reference image quality assessment algorithm. Most of
these assessment methods are developed on the basis of PSNR and SSIM. However, these met-
rics can neither calculate the structural component changes, nor tell the different types of dis-
tortions [9]. Dosselmann and Yang [10] demonstrated that a huge gap exists between the
PSNR or SSIM score and human perception. The results on subjective assessment databases
also show that these algorithms have a certain improvement in performance compared with
the classical metrics, but the overall performance is still unsatisfactory.

With regard to the no-reference image quality assessment method, Kim et al. [11, 12] pro-
posed a model based on generative adversarial networks, which was divided into prediction
and guidance. The image quality score is predicted according to the image content informa-
tion and location information, and the guidance was used to judge whether the scoring is man-
ual score or the output score of prediction network. These two parts are adversarial, which can
iteratively improve the performance of prediction network. In fact, the participators watch the
panoramic images from the perspective of viewports, instead of small blocks. Li et al. [13] pro-
posed a view based convolutional neural network (V-CNN) method. This method first used
CNN and non-maximum suppression to calculate the importance weight of the view, then
projected the view image to the plane. Subsequently, the method calculated the quality scores
of saliency map and view, and obtained the final quality score by weighting. Although the
method proposed by Yang et al. [14] fuses multi-level saliency map features based on weighted
PSNR and used neural network, the network can only learn the mapping relationship between
weighted PSNR and mean opinion score (MOS) in the training process, which is unstable and
difficult for the network to learn. Deep learning can extract many high-dimensional features
for quality evaluation, which makes its performance generally better than traditional methods.
However, deep learning-based methods depend on the quality score of the image patches, and
cannot globally perceive the quality of the panoramic image.

The main problem of existing methods is that they do not consider the global characteristics
of panoramic images, leading to the degradation of quality assessment precision. With regard
to the no-reference panoramic image quality assessment, the performance is largely dependent
on the feature extraction. In other words, the extracted features, which can globally reflect the
distortion degrees, are suitable for panoramic image quality assessment. By studying the statis-
tical characteristics of panoramic images, we summarize that the features should have three
characteristics: 1) The features should have low correlation to the content of the image. In no-
reference task, if the correlation between feature and image content is low, the distorted image
content does not affect the feature used for assessment. Therefore, the performance of the
assessment with such a low correlation model will be robust; 2) The features should be closely
related to the levels of distortion. In this way, the features can precisely reflect the distortion
without requirement of reference; 3) The feature size should be as small as possible. The size of
the feature has a great influence on the no-reference assessment performance. Actually, when
the feature dimension is much larger than the number of samples, the no-reference perfor-
mance is pretty poor.

Based on the observation above, we judge the degree of panoramic image distortion
through the correlation of adjacent pixels, and propose a no-reference quality assessment
method based on the multi-region adjacent pixels correlation (MRAPC) features. Our

proposed method can calculate the image quality score globally and achieve better performance. The contributions of this paper are as follows:

1. This paper analyzes the statistical characteristics of panoramic images, and further proves that the panoramic images distortion is highly related to the correlation of adjacent pixels. Therefore, the feature that can globally express the correlation between adjacent pixels is very suitable for quality assessment of coding distortion;

2. This paper proposes a no-reference panoramic image quality assessment method based on the correlation of adjacent pixels in multi-region. In this way, the global characteristics are considered to promote the objective quality assessment;

3. This paper reduces the feature dimension of panoramic image through shrinking the pixel value range, leading to the significantly computational efficiency improvement.

The rest of this paper is organized as follows. The overview of efficient feature extraction for panoramic images is presented in Section 2. Section 3 describes the proposed panoramic image quality assessment algorithm. Section 4 shows and analyzes the experimental results. Finally, the conclusions are drawn in Section 5.

## 2 The efficient feature for panoramic images

To obtain the expected features mentioned above, we conduct extensive experiments on panoramic images to study the statistical characteristics of their pixels based on the observation earlier. Fig 1 shows the probability distribution of adjacent pixel pairs of 91 panoramic images

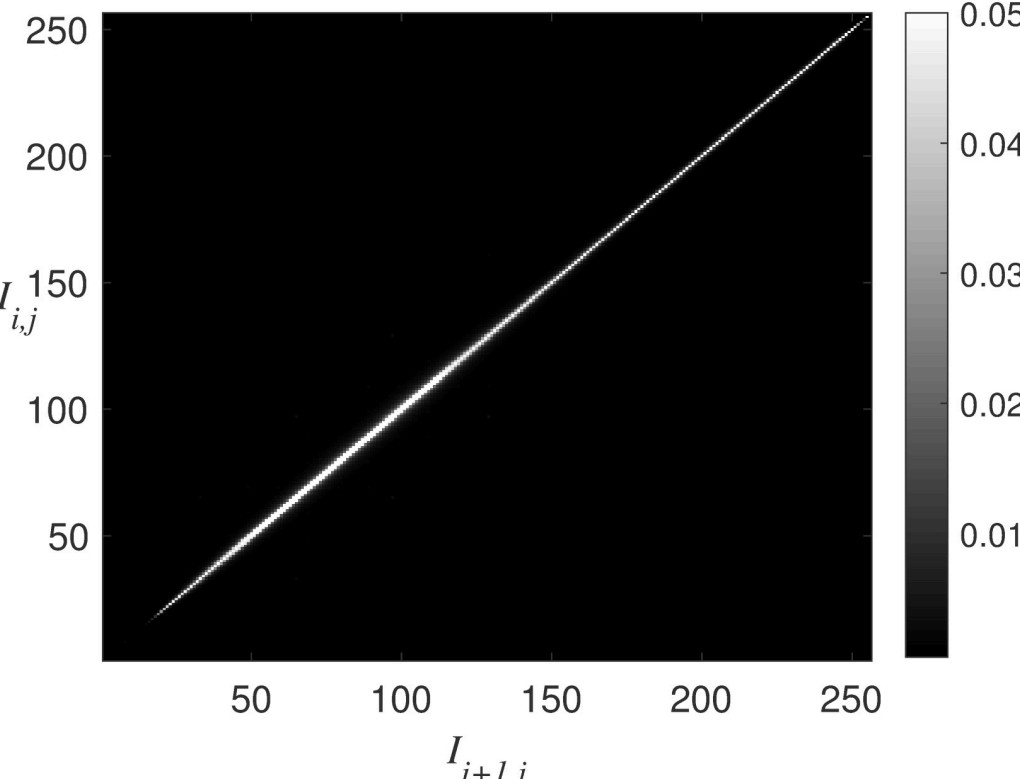

**Fig 1. The probability distribution of adjacent pixel pair ($I_{i,j}$, $I_{i+1,j}$).**

**Table 1. Six difference operators.** They are divided into horizontal and vertical operators, and each directional set consists of first-order, second-order and third-order operator.

| Direction | First-order operator | Second-order operator | Third-order operator |
|---|---|---|---|
| horizontal | $[-1, 1]$ | $[1, -2, 1]$ | $[1, -3, 3, -1]$ |
| vertical | $[-1, 1]^T$ | $[1, -2, 1]^T$ | $[1, -3, 3, -1]^T$ |

in CVIQD [15], OIQA [16] and VQA-ODV [17] datasets, where $I_{i,j}$ represents the pixel value of the panoramic image at coordinate $(i, j)$. It can be seen from Fig 1 that the values of adjacent pixels in panoramic images are very close, which reveals that the energy in the panoramic image is mainly concentrated in the low frequencies.

To this end, this paper analyzes the proportion of high-frequency information in the panoramic image by using various difference operators. It can be regarded as a high-pass filter, which can suppress the low-frequency information and display the high-frequency information in panoramic image. Taking the horizontal difference operator $[-1, 1]$ (see Table 1) as an example, it is the difference of the adjacent pixel in panoramic image, i.e., $I_{i,j} - I_{i+1,j}$. Fig 2 shows the probability distributions with the difference results under $I_{i,j} \in \{32, 64, 128, 192\}$. As can be seen from the Fig 2, more than 90% of the difference results are 0 and ±1. These probability distributions are similar to Gaussian distribution, which further indicates that the difference result is independent of the pixel value of panoramic image. In addition, because the difference result is very concentrated, we only need to consider the pixel values in a very small range and thereby further reducing the dimension of features.

We utilize the Pearson coefficient to calculate the correlations of adjacent pixels along horizontal direction, vertical direction and diagonal direction, respectively. It is found from Fig 3 that the correlation of adjacent pixels along diagonal direction is weaker than that along horizontal and vertical directions. Furthermore, Fig 4 depicts the probability distributions of adjacent pixel difference under six types of different distortions, in which each type of distortion contains three levels. It can be seen that the statistical characteristics vary with the change of distortion level.

It can be seen from the above experiments that the statistical characteristics of difference results are not only independent of panoramic image content, but also related to the degree of distortion. Consequently, the statistical characteristic is the expected indicator for measuring the distortion degree of the panoramic image.

## 3 Proposed method

In this section, we propose a no-reference quality assessment method based on multi-region adjacent pixel correlation (MRAPC) feature. The proposed method consists of four parts: residual calculation, residual truncation, co-occurrence matrix calculation and MRAPC-based Support Vector Regression (SVR) model. The framework is shown in Fig 5. Specifically, the MRAPC features are calculated by co-occurrence matrix, and then are fed into SVR for training and testing, and finally the objective quality assessment scores of panoramic images are obtained.

### 3.1 Calculation of difference map and threshold

To make the statistical characteristics more compact, the difference map is adopted in this paper to limit the image content into a small range, leading to the dimension reduction of the features. Specifically, this paper uses difference map to represent the pixel gradient. For an

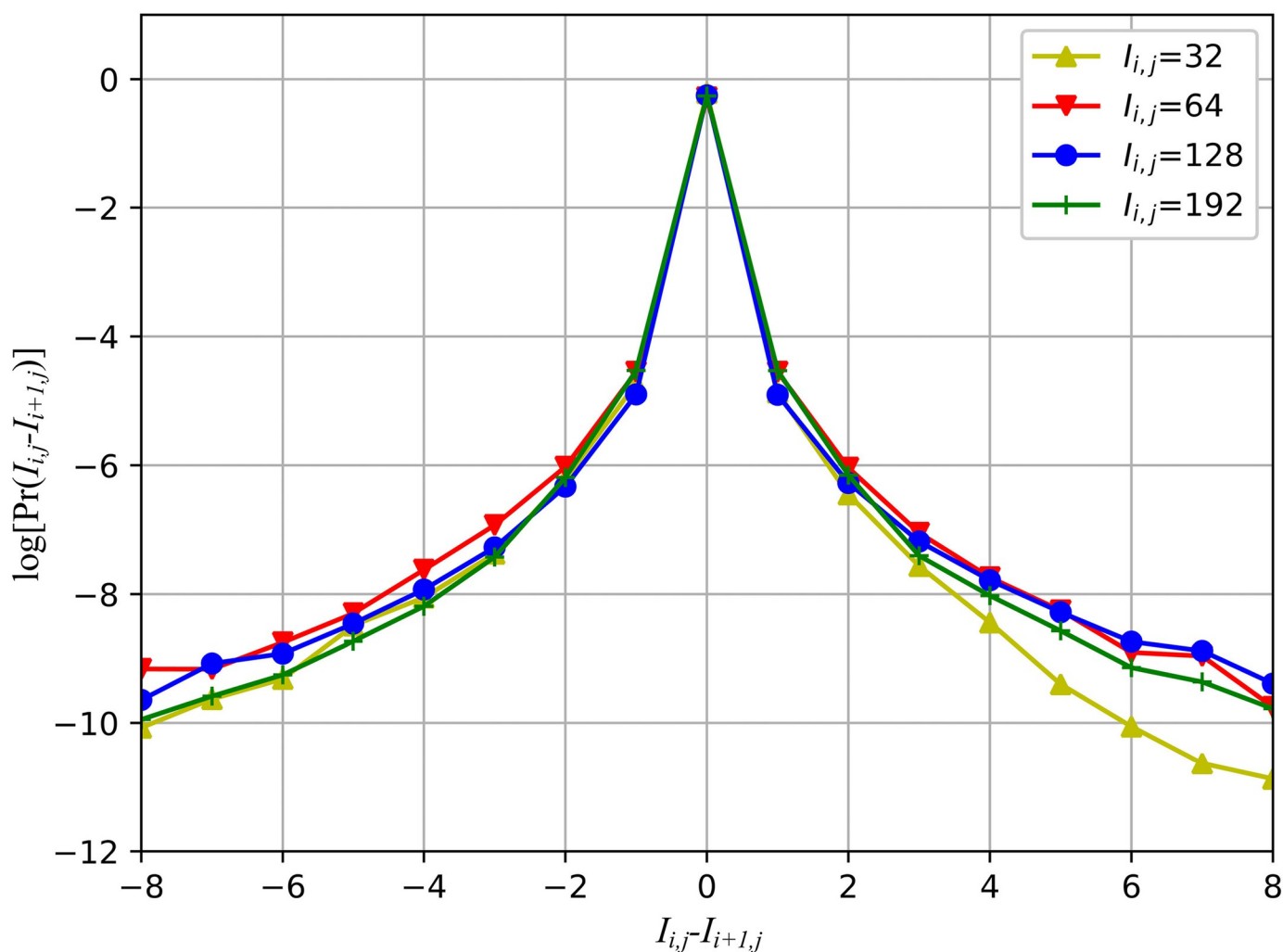

**Fig 2. The probability distributions of pixel difference $I_{i,j} - I_{i+1,j}$.** $Pr(I_{i,j} - I_{i+1,j})$ for $I_{i,j} \in \{32, 64, 128, 192\}$ estimated from 91 original panoramic images in CVIQA, OIQA and VQA-ODV databases.

image with size of $m \times n$, its $k$-order difference map is expressed as $\mathbf{R}^k = \bigcup_{i=1}^{M} \bigcup_{j=1}^{N} R_{i,j}^k$, where $R_{i,j}^k$ $k$-order difference value of position $(i, j)$.

In the practical calculation process, six kinds of difference operators (see Table 1) are used to obtain six kinds of residual maps. Various difference operators indicate the corresponding correlation of adjacent pixels. Taking the horizontal direction as an example, the first-order residual is calculated by $R_{i,j}^1 = I_{i,j+1} - I_{i,j}$, the second-order residual by $R_{i,j}^2 = I_{i,j+1} - 2I_{i,j} + I_{i,j-1}$, and the third-order residual by $R_{i,j}^3 = I_{i,j-1} - 3I_{i,j} + 3I_{i,j+1} - I_{i,j+2}$. The calculation along the vertical direction is similar as that along horizontal direction.

It should be noted that since most of the residual values are concentrated in a small range, we set a threshold $T = 2$ for difference map $\mathbf{R}$ to reduce the residual range:

$$
\begin{cases}
R_{i,j} = -T, R_{i,j} < -T \\
R_{i,j} = T, R_{i,j} > T.
\end{cases}
\tag{1}
$$

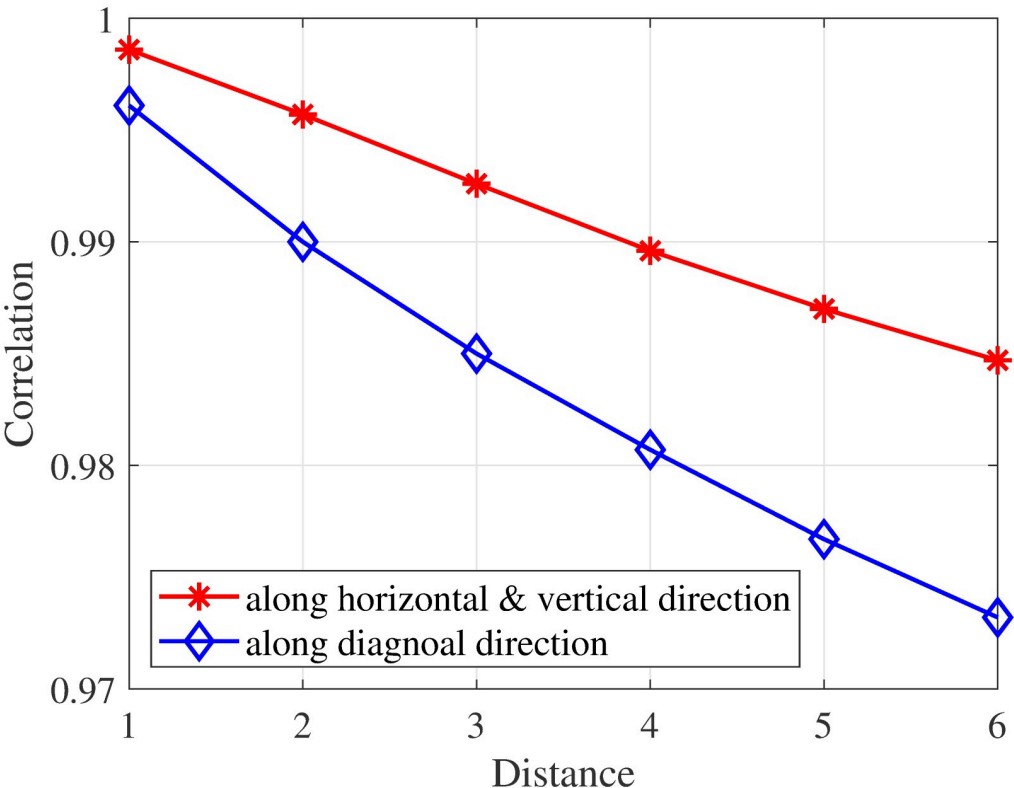

**Fig 3. Correlation between adjacent pixels with pixel distance in panoramic image.** The red line represents the adjacent pixels along horizontal and vertical directions, and the blue line represents the adjacent pixels along diagonal direction.

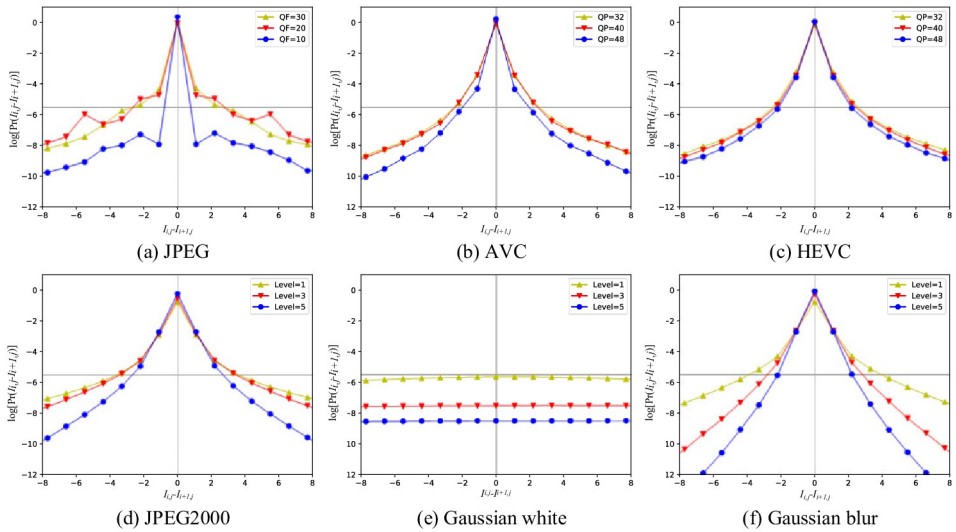

**Fig 4. The probability distribution of pixel difference $I_{i,j} - I_{i+1,j}$ under different distortion levels of each distortion type.** (a) JPEG. (b) AVC. (c) HEVC. (d) JPEG2000. (e) Gaussian white. (f) Gaussian blue.

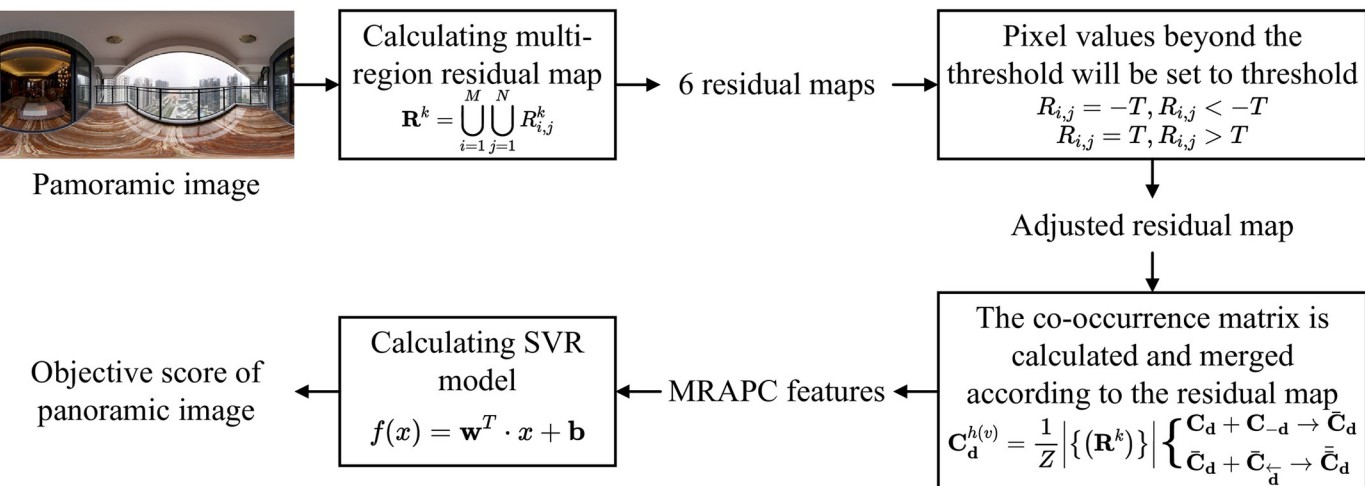

**Fig 5. Framework of quality assessment method based on multi-region adjacent pixel correlation feature.**

### 3.2 Calculation of co-occurrence matrix

In this section, we use the fourth-order co-occurrence matrix to describe the pixel gradient in residual map. As mentioned earlier, the correlation of pixels along diagonal direction is weaker than that along horizontal and vertical directions, so for simplicity, the pixel gradient is only calculated along horizontal and vertical directions. Taking the horizontal direction as an example, a four-dimensional hypercube $\mathbf{C}_{\mathbf{d}}^{h}$, which is generated by a fourth-order co-occurrence matrix, can represent the joint probability distribution of four adjacent pixels along the horizontal direction:

$$\mathbf{C}_{\mathbf{d}}^{h} = \frac{1}{Z}|\{\left(R_{i,j}, R_{i,j+1}, R_{i,j+2}, R_{i,j+3}\right)|R_{i,j+k-1} = d_k, k = 1, \cdots, 4\}|, \tag{2}$$

where $\mathbf{d} = (d_1, d_2, d_3, d_4) \in \{-2, -1, 0, 1, 2\}^4$ and $Z$ is the normalization coefficient, which makes $\sum_{\mathbf{d} \in \tau_4} \mathbf{C}_{\mathbf{d}} = 1$ Take $\mathbf{d}^* = (0, 0, 0, 0)$ as an example, the value of $\mathbf{C}_{\mathbf{d}^*}^{h}$ is the joint probabilities that residual point $R_{i,j}$ and its horizontal adjacent points $R_{i+1,j}, R_{i+2,j}, R_{i+3,j}$ are all equal to 0. The calculation of $\mathbf{C}_{\mathbf{d}}^{v}$ along vertical direction is similar as that along horizontal direction, i.e., the joint probability distribution of vertical adjacent points $R_{i,j}, R_{i+1,j}, R_{i+2,j}, R_{i+3,j}$. Note that each difference map with the same order corresponds to two directional co-occurrence matrices.

### 3.3 Symmetric integration of co-occurrence matrix

The four-dimensional hypercube $\mathbf{C}_{\mathbf{d}}$ has $|\tau_4| = 625$ elements in total, leading to a lot of redundancy. For efficiency, we propose to symmetrically integrate the elements in each co-occurrence matrix according to the following two rules:

$$\mathbf{C}_{\mathbf{d}} + \mathbf{C}_{-\mathbf{d}} \rightarrow \bar{\mathbf{C}}_{\mathbf{d}}, \tag{3}$$

$$\bar{\mathbf{C}}_{\mathbf{d}} + \bar{\mathbf{C}}_{\overleftarrow{\mathbf{d}}} \rightarrow \bar{\bar{\mathbf{C}}}_{\mathbf{d}}, \tag{4}$$

where $-\mathbf{d} = (-d_1, -d_2, -d_3, -d_4)$, $\overleftarrow{\mathbf{d}} = (d_4, d_3, d_2, d_1)$. The opposite subscript $-\mathbf{d}$ only changes the monotonicity of the pixel values, but remains the pixel gradient; The reverse subscript $\overleftarrow{\mathbf{d}}$

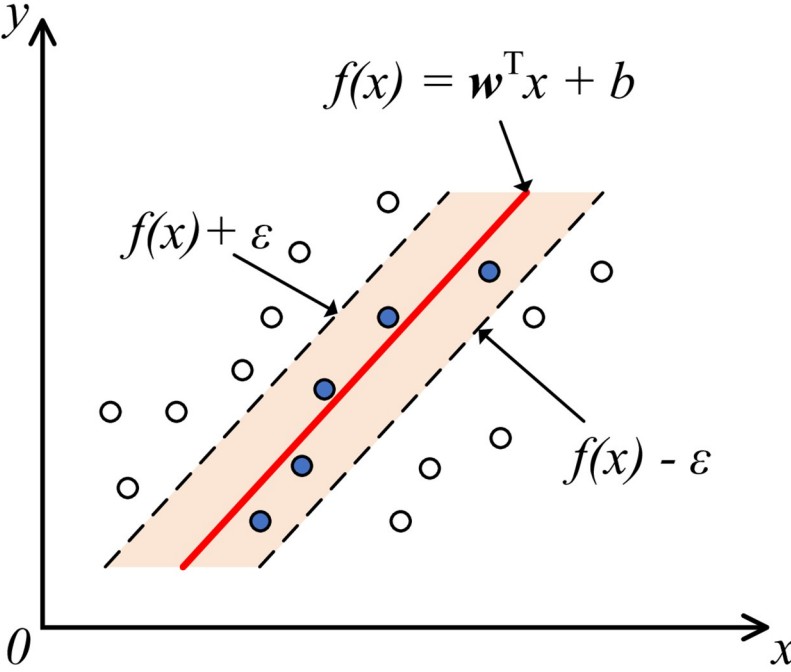

**Fig 6. Support vector regression diagram.**

does not affect the pixel gradient either. After integrating, the number of elements in the co-occurrence matrix is reduced from 625 to 169. The MRAPC feature is computed by six difference operators and thus has $6 \times 169 = 1014$ dimensions, which is lightweight. Due to the preservation of pixel gradient, the symmetric integration greatly reduces the dimension of features while keeping the effectiveness of features.

### 3.4 SVR model training and testing

SVR is a widely used form of support vector machine [18] to solve regression problems, so we adopt SVR to get the final quality score of panoramic images. As shown in Fig 6, for a given sample $D = \{(x_1, y_1), (x_2, y_2), \cdots, (x_n, y_n)\}$, $y_i \in \mathbf{R}$, we construct a regression model in the form of $f(x) = \mathbf{w}^T \cdot x + \mathbf{b}$, to make $f(x)$ and $y$ as close as possible, where $\mathbf{w}$ and $\mathbf{b}$ are model parameters. Since it is a regression problem that needs to be calculated, the model can tolerate a deviation of at most $\varepsilon$ between $f(x)$ and $y$. We divide the database into training set and test set. Then, we feed the MRAPC features (assigned as $x$) of panoramic images of training set and corresponding MOS (assigned as $y$) into SVR for training to get the trained SVR model. Finally, the features of the test images are considered to predict the quality scores.

## 4 Experimental results

### 4.1 Experimental settings

For experiments, the assessment algorithm with the proposed MRAPC feature is tested on the CVIQD [15] and OIQA [16] databases, due to their rich distortions for the panoramic images. The CVIQD contains 16 reference panoramic images and each image corresponds to 33 distorted images, which have three distortion types, namely JPEG coding distortion, AVC coding distortion and HEVC coding distortion. The quality factor of JPEG ranges from 0 to 50, and

the interval is 5; The quantization parameters of the AVC and HEVC range from 30 to 50, with an interval of 2. The OIQA consists of 16 reference panoramic images and each image corresponds to 20 distorted images, which have four distortion types, namely JPEG coding distortion, JPEG2000 coding distortion, Gaussian blur and Gaussian white noise.

Note that the proposed algorithm follows the recommendations given in [19] and uses a 5-parameter logistic function to fit the predicted results, as shown in Eq (5).

$$f(x) = \beta_1 \left( \frac{1}{2} - \frac{1}{1 + e^{\beta_2(x-\beta_3)}} \right) + \beta_4 x + \beta_5, \tag{5}$$

where $x$ denotes the score predicted by the proposed algorithm, $\{\beta_i | i = 1, 2, \cdots, 5\}$ are the five parameters to be fitted, and $f(x)$ denotes the final quality score. We use three commonly used evaluation metrics: Spearman Rank-order Correlation Coefficient (SROCC), Pearson Linear Correlation Coefficient (PLCC) and Root Mean Squared Error (RMSE) to evaluate the proposed algorithm. 1) Spearman Rank-order Correlation Coefficient is calculated by:

$$SROCC = 1 - \frac{6 \sum_{i=1}^{N} d_i^2}{N(N^2 - 1)}, \tag{6}$$

where $N$ is the number of images and $d_i$ is the difference between the subjective score and the objective score of the $i$-th image. 2) Pearson Linear Correlation Coefficient is computed by:

$$PLCC = \frac{\sum_{i=1}^{N} (s_i - \mu_s)(o_i - \mu_o)}{\sqrt{\sum_{i=1}^{N} (s_i - \mu_s)^2 \times \sum_{i=1}^{N} (o_i - \mu_o)^2}}, \tag{7}$$

where $N$ is the number of images; $s_i$ and $o_i$ denote the subjective and objective assessment score of the $i$-th image; $\mu_s$ and $\mu_o$ represent the corresponding mean values of $s_i$ and $o_i$, respectively. 3) Root Mean Squared Error is calculated by:

$$RMSE = \sqrt{\frac{\sum_{i=1}^{N} (s_i - o_i)^2}{N}}, \tag{8}$$

where $N$ is the number of images; $s_i$ and $o_i$ denote the subjective and objective assessment score of the $i$-th image.

These evaluation metrics can reflect assessment algorithms performance from different aspects. Specifically, SROCC mainly focuses on the monotonicity of prediction, as well as PLCC and RMSE on the accuracy of prediction. Note that stronger correlation and lower error indicate higher performance.

Both CVIQD and OIQA databases contain 16 scenes. In order to ensure the reliability of the results, the images of 12 scenes are randomly selected for training. The images of the remaining 4 scenes are selected for testing. After repeating 1000 cross validations, the median of obtained SROCC scores and the other metrics in all experiments are taken as the final experimental results. In the experiment, the SVM function "fitrsvm" of MATLAB 2018a is selected for training, where the radial basis function (RBF) is selected as the kernel function and other parameters keep the default configurations.

## 4.2 Overall performance analysis

In order to verify the advantages of the proposed method, this section compares the proposed algorithm with the existing static image assessment algorithms SSIM [3], BRISQUE [8], and the panoramic image quality assessment algorithms S-PSNR [4], WS-PSNR [5], CPP-PSNR [6], WS-SSIM [7]. The experimental results are shown in Table 2. From the results, we can

**Table 2. Comparison of the overall performance of the algorithm with the proposed MRAPC and other panoramic image quality assessment algorithms.**

| Database | CVIQD | | | OIQA | | |
|---|---|---|---|---|---|---|
| Metrics | SROCC↑ | PLCC↑ | RMSE↓ | SROCC↑ | PLCC↑ | RMSE↓ |
| S-PSNR [4] | 0.8759 | 0.8872 | 6.6382 | 0.4533 | 0.4901 | 1.8906 |
| WS-PSNR [5] | 0.8760 | 0.8887 | 6.5988 | 0.4705 | 0.5309 | 1.8383 |
| CPP-PSNR [6] | 0.8734 | 0.8869 | 6.6462 | 0.4623 | 0.4891 | 1.8918 |
| SSIM [3] | 0.8025 | 0.8312 | 8.0013 | 0.3743 | 0.2577 | 2.0956 |
| WS-SSIM [7] | 0.8063 | 0.8344 | 7.9306 | 0.3829 | 0.2837 | 2.0798 |
| BRISQUE [8] | 0.8817 | 0.9156 | 5.9067 | 0.9142 | 0.9222 | 0.8507 |
| MRAPC | **0.9461** | **0.9616** | **3.7392** | **0.9419** | **0.9469** | **0.6914** |

conclude the following three points. 1) The PSNR-based algorithms are conducted at the pixel level, so it fails to consider the correlation between pixels. As illustrated in Table 2, the highest SROCC for these algorithms is 0.8760 of WS-PSNR; 2) The SSIM-based algorithms consider the image structure, but the nonlinear structural distortion caused by projection has a great influence on the calculation of SSIM. WS-SSIM, as the best SSIM-based algorithms, is still insufficiently effective; 3) The proposed algorithm can globally judge the degree of distortion according to the correlation of adjacent pixels, so the highest performance is achieved. In a word, the algorithm with the proposed MRAPC is superior to the other panoramic image quality assessment algorithms.

In addition, due to the richer scenes in the CVIQD, we select it to intuitively illustrate algorithm performance. Fig 7 shows the scatter diagrams and fitting curves of different algorithms on CVIQD. Since scatter diagrams and fitting curves reflect subjective and objective quality scores, respectively, it means the algorithm performance higher that more scatter points are located on the fitting curve and more uniform of the points exist beside the curve. As can be seen from Fig 7, the quality scores predicted by the proposed algorithms have a stronger correlation with MOS scores.

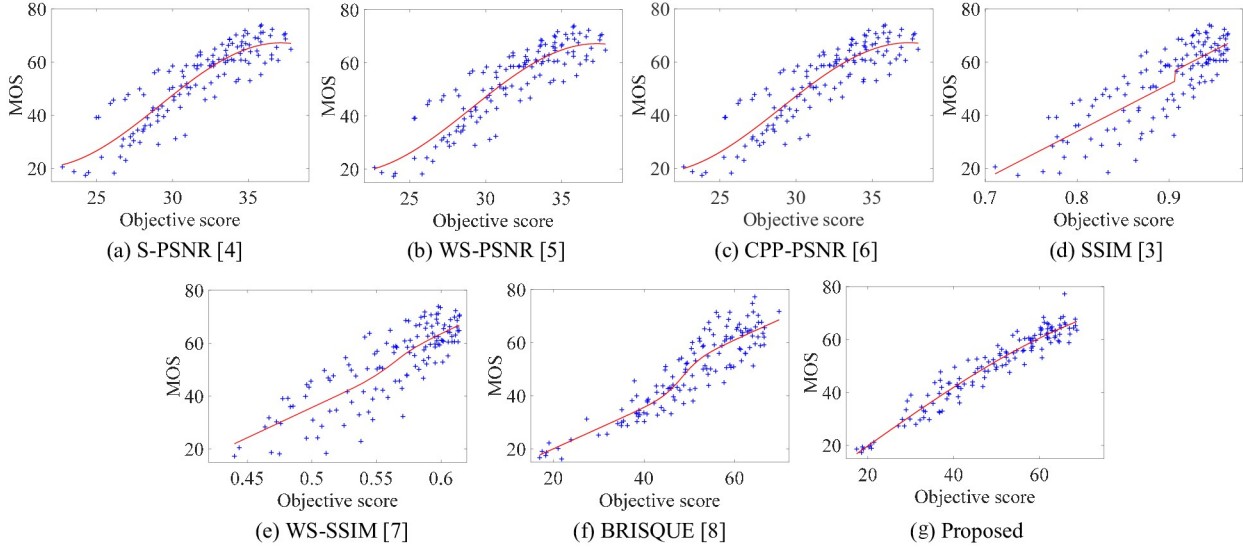

(a) S-PSNR [4]    (b) WS-PSNR [5]    (c) CPP-PSNR [6]    (d) SSIM [3]

(e) WS-SSIM [7]    (f) BRISQUE [8]    (g) Proposed

**Fig 7. Scatter diagrams for the pairs of MOS score and objective score of different algorithms.** (a) S-PSNR [4]. (b) WS-PSNR [5]. (c) CPP-PSNR [6]. (d) SSIM [3]. (e) WS-SSIM [7]. (f) BRISQUE [8]. (g) Proposed.

**Table 3. Performance comparison of methods under different distortion types.**

| Distortion Type | JPEG | | | AVC | | | HEVC | | |
|---|---|---|---|---|---|---|---|---|---|
| Metrics | SROCC↑ | PLCC↑ | RMSE↓ | SROCC↑ | PLCC↑ | RMSE↓ | SROCC↑ | PLCC↑ | RMSE↓ |
| S-PSNR [4] | 0.8403 | 0.9271 | 6.3303 | 0.9236 | 0.8920 | 5.8033 | 0.9121 | 0.8876 | 5.8679 |
| WS-PSNR [5] | 0.8307 | 0.9213 | 6.5663 | 0.9043 | 0.8882 | 5.8978 | 0.9051 | 0.8776 | 6.1089 |
| CPP-PSNR [6] | 0.8297 | 0.9193 | 6.6453 | 0.9026 | 0.8837 | 6.0099 | 0.9071 | 0.8749 | 6.1710 |
| SSIM [3] | 0.7737 | 0.8941 | 7.5629 | 0.8080 | 0.8161 | 7.4190 | 0.8321 | 0.8295 | 7.7157 |
| WS-SSIM [7] | 0.7801 | 0.8992 | 7.3874 | 0.8321 | 0.8462 | 6.8405 | 0.8465 | 0.8497 | 6.7175 |
| BRISQUE [8] | 0.8802 | 0.9069 | 5.8788 | 0.9220 | **0.9686** | 4.2218 | 0.9072 | 0.9105 | 5.1460 |
| MRAPC | **0.9522** | **0.9830** | **3.1336** | **0.9529** | 0.9602 | **3.2341** | **0.9285** | **0.9437** | **3.9551** |

## 4.3 Performance analysis of different distortion types

To further demonstrate the accuracy of the proposed algorithm in terms of coding distortions, we additionally conduct extensive experiments on the three kinds of compression distortions of CVIQD. In Table 3, the result show that our proposed algorithm is comprehensively better than the other algorithms in terms of the distortions by JPEG, AVC, and HEVC. With regard to the AVC distortion, the PLCC score of our algorithm is slightly lower than that of the BRIS-QUE algorithm, but the difference 0.0084 is negligible. It proves that our proposed algorithm has potential to guide the improvement on panoramic image compression.

## 4.4 Ablation experiment

The ablation experiment is carried out to verify the effectiveness of the three residual calculation methods. Table 4 illustrates the results of the "first-order" MRAPC features, the "first-order + second-order" MRAPC features, and the "first-order + second-order + third-order" MRAPC features, respectively. It can be seen from Table 4 that with the introduction of higher order difference operator, the performance is improved. It verifies the effectiveness of the proposed multi-region adjacent pixels correlation feature.

## 4.5 Model verification experiment

In order to verify the generalization ability of our algorithm, we use the JPEG distorted images in OIQA or those in CVIQD, respectively, as the training set or the test set. Table 5 shows the results in these two experimental cases. In both cases, the proposed algorithm achieves

**Table 4. Performance comparison of MRAPC method with different residual orders.**

| Databases | CVIQD | | | OIQA | | |
|---|---|---|---|---|---|---|
| Metrics | SROCC↑ | PLCC↑ | RMSE↓ | SROCC↑ | PLCC↑ | RMSE↓ |
| first-order | 0.9390 | 0.9535 | 4.1801 | 0.9366 | 0.9402 | 0.7632 |
| first-order + second-order | 0.9409 | 0.9588 | 4.0384 | 0.9382 | 0.9466 | 0.6961 |
| first-order + second-order + third-order | **0.9461** | **0.9616** | **3.7392** | **0.9419** | **0.9469** | **0.6914** |

**Table 5. Performance comparison of MRAPC method with different residual orders.**

| Train Set | Test Set | SROCC↑ | PLCC↑ | RMSE↓ |
|---|---|---|---|---|
| OIQA_JPEG | CVIQD_JPEG | 0.9061 | 0.7471 | 5.5143 |
| CVIQD_JPEG | OIQA_JPEG | 0.9312 | 0.9439 | 0.7725 |

relatively high correlation and accuracy, which proves the generalization ability of our proposed algorithm is outstanding.

## 5 Conclusion

This paper aims at proposing an efficient no-reference objective quality assessment for panoramic images. We comprehensively analyze the property of panoramic images, and find that the adjacent pixels correlation of panoramic image is not only highly closed to the statistical characteristics of the image, but also prone to be affected by distortions. Based on this observation, we propose a no-reference quality assessment method of panoramic images based on the multi-region adjacent pixels correlation in this paper. Firstly, the different map of panoramic image is calculated by using the first-order, the second-order and the third-order difference operators, respectively. Then, the difference map is truncated, and the joint probability distribution of four adjacent pixels on the residual is calculated by using the fourth-order co-occurrence matrix. The probability distribution is input into SVR model as a feature, and the final quality assessment score is obtained. The experimental results show that our algorithm has higher performance than state-of-the-art algorithms.

## Author Contributions

**Formal analysis:** Xinpeng Huang.

**Funding acquisition:** Xinpeng Huang, Ping An.

**Investigation:** Wenxin Ding, Chunli Meng.

**Methodology:** Wenxin Ding.

**Resources:** Xin Liu.

**Supervision:** Xinpeng Huang, Ping An.

**Validation:** Chunli Meng.

**Writing – original draft:** Xinpeng Huang.

**Writing – review & editing:** Xinpeng Huang, Xin Liu, Wenxin Ding, Ping An.

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
