## [Decision Letter · Decision Letter 0]

7 Feb 2022

PONE-D-21-39405No-reference panoramic image quality assessment based on multi-region adjacent pixels correlationPLOS ONE

Dear Dr. An,

Thank you for submitting your manuscript to PLOS ONE. After careful consideration, we feel that it has merit but does not fully meet PLOS ONE’s publication criteria as it currently stands. Therefore, we invite you to submit a revised version of the manuscript that addresses the points raised during the review process.

We look forward to receiving your revised manuscript.

Kind regards,

Zhaoqing Pan, Ph.D.

Academic Editor

PLOS ONE

Journal Requirements:

 [This work was supported in part by the National Natural Science Foundation of China [Grants 62020106011, 62071287, 61901252, 62001279], and Science and Technology Commission of Shanghai Municipality [Grant 20DZ2290100].]

[This work was supported in part by the National Natural Science Foundation of China 

[Grants 62020106011, 62071287, 61901252, 62001279], and Science and Technology Commission of Shanghai Municipality [Grant 20DZ2290100].]

 [This work was supported in part by the National Natural Science Foundation of China [Grants 62020106011, 62071287, 61901252, 62001279], and Science and Technology Commission of Shanghai Municipality [Grant 20DZ2290100].]

Reviewers' comments:

Reviewer's Responses to Questions

**Comments to the Author**

1. Is the manuscript technically sound, and do the data support the conclusions?

Reviewer #1: Yes

Reviewer #2: Yes

2. Has the statistical analysis been performed appropriately and rigorously? 

Reviewer #1: Yes

Reviewer #2: Yes

3. Have the authors made all data underlying the findings in their manuscript fully available?

Reviewer #1: Yes

Reviewer #2: Yes

4. Is the manuscript presented in an intelligible fashion and written in standard English?

Reviewer #1: Yes

Reviewer #2: Yes

5. Review Comments to the Author

Reviewer #1: In this paper, a novel statistical feature is proposed for panoramic image quality assessment, and its validity is fully demonstrated, and the method is introduced clearly. Generally, this work is interesting and reasonable. However, this paper can be improved from the following aspects:

-In Section 3, the proposed method is presented in four parts, for each of them please use mathematical formulas to summarize the processing.

-In Eq.5, the authors propose a nonlinear mapping of the predicted scores; please explain the reasons for this approach here.

Reviewer #2: PONE-D-21-39405

The author proposed an novel panoramic image quality assessment based on multi-region adjacent pixels correlation to address the problem that the quality of panoramic image cannot be fully reflected due to the local division of panoramic image. The motivation is sufficient, and the method is described in detail. At the same time, a series of ablation and comparative experiments are conducted to verify the effectiveness of this method. Finally, please refer to the following revision for modification and explanation.

Minor Revision:

The author should clarify the different between the panoramic image and the normal image. Specifically, the reviewer suggests that the Introduction (line 2-7) can be rewritten in detail.

The font of I_{i+1,j} in Fig 1 is different from other font in this manuscript.

The author should clarify the parameter of the SVR such as "Penalty coefficient".

The size of Fig 7 should be enlarged.

6. PLOS authors have the option to publish the peer review history of their article (what does this mean?). If published, this will include your full peer review and any attached files.

Reviewer #1: No

Reviewer #2: No

---

## [Author Response · Author response to Decision Letter 0]

21 Feb 2022

We thank the editor and the reviewers for their insightful comments, and we have revised the manuscript based on their suggestions. Those comments are all valuable and very helpful for revising and improving our manuscript, as well as providing important guiding significance to our research. We have studied the comments carefully and have made the corrections, and we hope meet with approval. The point-by-point replies are given as the attachment.

---

## [Decision Letter · Decision Letter 1]

14 Mar 2022

No-reference panoramic image quality assessment based on multi-region adjacent pixels correlation

PONE-D-21-39405R1

Dear Dr. An,

We’re pleased to inform you that your manuscript has been judged scientifically suitable for publication and will be formally accepted for publication once it meets all outstanding technical requirements.

Kind regards,

Zhaoqing Pan, Ph.D.

Academic Editor

PLOS ONE

Additional Editor Comments (optional):

Reviewers' comments:

Reviewer's Responses to Questions

**Comments to the Author**

1. If the authors have adequately addressed your comments raised in a previous round of review and you feel that this manuscript is now acceptable for publication, you may indicate that here to bypass the “Comments to the Author” section, enter your conflict of interest statement in the “Confidential to Editor” section, and submit your "Accept" recommendation.

Reviewer #1: All comments have been addressed

Reviewer #2: All comments have been addressed

2. Is the manuscript technically sound, and do the data support the conclusions?

Reviewer #1: Yes

Reviewer #2: Yes

3. Has the statistical analysis been performed appropriately and rigorously? 

Reviewer #1: Yes

Reviewer #2: Yes

4. Have the authors made all data underlying the findings in their manuscript fully available?

Reviewer #1: Yes

Reviewer #2: Yes

5. Is the manuscript presented in an intelligible fashion and written in standard English?

Reviewer #1: Yes

Reviewer #2: Yes

6. Review Comments to the Author

Reviewer #1: (No Response)

Reviewer #2: (No Response)

7. PLOS authors have the option to publish the peer review history of their article (what does this mean?). If published, this will include your full peer review and any attached files.

Reviewer #1: No

Reviewer #2: No

---

## [Editor Report · Acceptance letter]

18 Mar 2022

PONE-D-21-39405R1 

No-reference panoramic image quality assessment based on
 multi-region adjacent pixels correlation 

Dear Dr. An:

I'm pleased to inform you that your manuscript has been deemed suitable for publication in PLOS ONE. Congratulations! Your manuscript is now with our production department. 

Kind regards, 

on behalf of

Dr. Zhaoqing Pan 

Academic Editor

PLOS ONE